# Reviving the Weizmann process for commercial n-butanol production

Ngoc-Phuong-Thao Nguyen[1,2,3,4], Céline Raynaud[5], Isabelle Meynial-Salles[1,2,3] & Philippe Soucaille[1,2,3,5,6]

Developing a commercial process for the biological production of n-butanol is challenging as it needs to combine high titer, yield, and productivities. Here we engineer *Clostridium acetobutylicum* to stably and continuously produce n-butanol on a mineral media with glucose as sole carbon source. We further design a continuous process for fermentation of high concentration glucose syrup using in situ extraction of alcohols by distillation under low pressure and high cell density cultures to increase the titer, yield, and productivity of n-butanol production to the level of 550 g/L, 0.35 g/g, and 14 g/L/hr, respectively. This process provides a mean to produce n-butanol at performance levels comparable to that of corn wet milling ethanol plants using yeast as a biocatalyst. It may hold the potential to be scaled-up at pilot and industrial levels for the commercial production of n-butanol.

[1] INSA, UPS, INP, LISBP, Université de Toulouse, Toulouse, France. [2] INRA, UMR792, Toulouse, France. [3] CNRS, UMR5504, Toulouse, France. [4] School of Medicine, Tan Tao University, Duc Hoa, Tan Duc e-City, Long An, Vietnam. [5] Metabolic Explorer, Biopôle Clermont-Limagne, Saint Beauzire, France. [6] BBSRC EPSRC Synthetic Biological Research Center SBRC, School of Life Sciences, University of Nottingham, Nottingham, England. Correspondence and requests for materials should be addressed to P.S. (email: soucaille@insa-toulouse.fr)

The Weizmann process was developed at the beginning of the 20th century for the biological conversion of corn into acetone and n-butanol by *Clostridium acetobutylicum*[1–3]. It was the second largest fermentation process (after ethanol) of enormous industrial, social and historical importance[1–3]. Beyond its use during the First World War to produce acetone for smokeless gunpowder (cordite) manufacturing, it became a process used worldwide to produce these two industrial solvents from a variety of renewable substrates[3–5]. Its demise in the early 1960s was the result of the superior petrochemical-process economics due to drastic increase of the substrate costs[1], the low yield, titer, and productivity of the fermentation[6] (resulting in high operating and capital expenses), and the inability to use continuous process technologies (as in the case of petrochemical processes) due to the loss of *C. acetobutylicum*'s capacity to produce solvents because of degeneration[7,8].

Here we first use an advanced metabolic engineering approach to engineer *C. acetobutylicum* to stably and continuously produce n-butanol from glucose[9]. We further design a continuous process for the fermentation of high concentration glucose syrup using (i) in situ extraction of alcohols by distillation under low pressure, and (ii) high cell density cultures to increase both the titer, yield, and productivity of n-butanol production.

## Results

**Engineering *C. acetobutylicum* for the production of *n*-butanol.** This step requires to first eliminate the production of undesired by-products namely butyrate, acetone, and lactate (Fig. 1a). In the CAB1057 strain, we have then deleted *ptb* and *buk* that code for the phosphotransbutyrylase and the butyrate kinase involved in the last two steps of butyrate formation[10–12], *ctfAB* that code for the acetoacetyl-CoA-acetate CoA-transferase involved in the first specific step of acetone formation[13] and *ldhA* that codes for the lactate dehydrogenase involved in the last step of L-lactate formation[13]. When CAB1057 was evaluated in chemostat culture and subjected to a metabolic flux analysis (Supplementary Fig. 1),

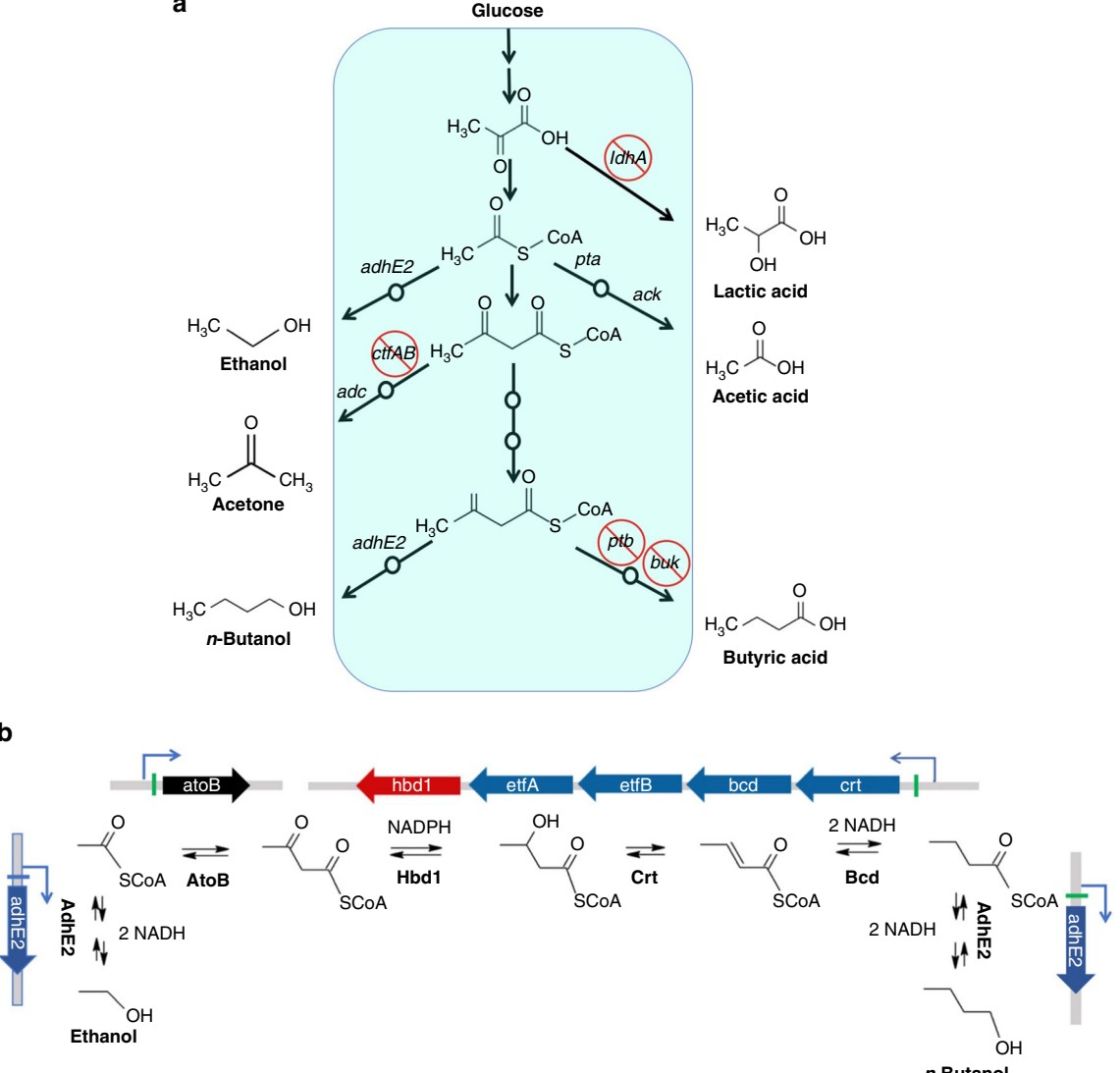

**Fig. 1** Metabolic engineering of *Clostridium acetobutylicum* for *n*-butanol production. **a** Elimination of by-products formation (strain CAB1057) by gene deletion. *ldhA* lactate dehydrogenase, *ctfAB* acetoacetyl-CoA-acetate CoA-transferase, *adc* acetoacetate decarboxylase, *ptb* phosphotransbutyrylase, *buk* butyrate kinase, *adhE2* bifunctional NAD+-dependent aldehyde and alcohol dehydrogenase. **b** Optimization of acetyl-CoA conversion to *n*-butanol (Strain CAB1060) using a chimeric pathway derived from three different microorganisms. Blue *C. acetobutylicum*; black *E. coli*, red *C. kluyveri*; Green boxes are Rex binding sites. *atoB* acetoacetyl-CoA thiolase/synthase, *hbd1* NADP+-dependent 3-hydroxybutyryl-CoA dehydrogenase, *crt* crotonase, *bcd-etfAB* butyryl-CoA dehydrogenase

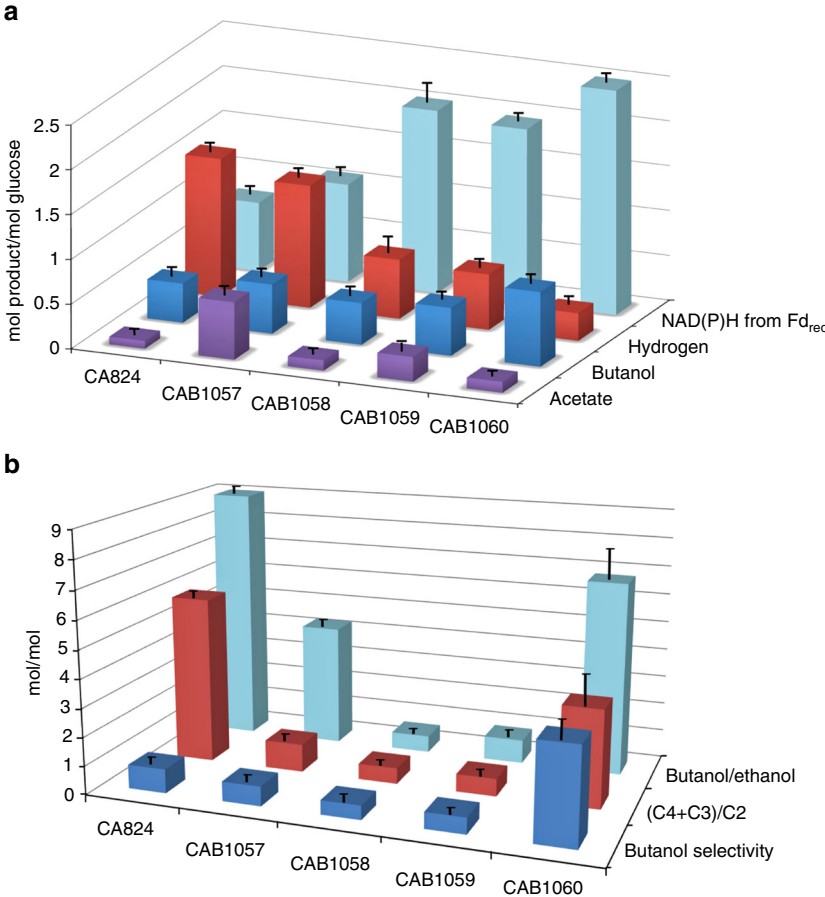

**Fig. 2** Optimization of the *n*-butanol-producing strain. All strains were evaluated in phosphate-limited chemostat cultures at pH 4.4 for wild type and 5.0 for all the other strains. All the engineered strain produced no acetone, lactate, and butyrate. **a** *n*-Butanol, acetate, hydrogen, and NAD(P)H from $Fd_{red}$ yields (mol/mol glucose). **b** Product ratios (mol/mol). (C4 + C3)/C2 = (Butanol + Butyric acid + Acetone)/ (Ethanol + Acetic acid). Butanol selectivity = Butanol/(Σ of all products except butanol). Error bars represent SD of three biological replicates

the expected phenotype was obtained for the deleted pathways but the *n*-butanol yield (Fig. 2a) and fluxes (Supplementary Fig. 1) were not improved as this mutant produces more acetate and ethanol than the wild-type strain. The redox-sensing transcriptional repressor (Rex) has been found to modulate its DNA-binding activity in response to the NADH/NAD$^+$ ratio and to repress the expression of several operons including *thlA* (that codes for a thiolase), *crt-bcd-etfAB-hbd*, (that code, respectively, for a crotonase, a butyryl-CoA dehydrogenase, and a 3-hydroxy-butyryl-CoA dehydrogenase) and *adhE2* (that codes for a bifunctional aldehyde-alcohol[14,15] (Fig. 1b)). As *thlA* and *crt-bcd-etfAB-hbd* code for the enzymes converting acetyl-CoA to butyryl-CoA and *adhE2* codes for the enzyme that reduces butyryl-CoA and acetyl-CoA to *n*-butanol and ethanol, respectively, the *rexA* gene, encoding Rex, was deleted in CAB1057 (to yield CAB1058) with the objective of improving the C4/C2 ratio and the flux of alcohol formation. The yield and flux of alcohol formation by CAB1058 were significantly improved, associated with a four-fold decrease of the hydrogen yield and fluxes as the electrons from reduced ferredoxin were mainly used for NADH formation (Fig. 2a, Supplementary Fig. 1). However, the higher alcohol yield was only due to an increase in the ethanol flux as the *n*-butanol yield remained almost unchanged and, consequently, the C4/C2 ratio was not improved (Fig. 2b). When the thiolase (catalyzed by ThlA) and 3-hydroxy-butyryl-CoA dehydrogenase (catalyzed by Hbd) activities were measured in CAB1057 and CAB1058, ThlA remained unchanged while Hbd was 1.5 fold higher), indicating that the *rexA* deletion has the expected effect

on the expression of the *crt-bcd-etfAB-hbd* operon coding for the enzymes that convert acetoacetyl-CoA to butyryl-CoA) but not on *thlA*. ThlA is one of the most abundant intracellular protein[16] representing more than 5% of the cytosolic proteins. Furthermore, the condensation reaction by ThlA was shown to be very sensitive to CoA-SH, with micro molar levels totally inhibiting the enzyme[17] and it was also found to be regulated by a redox-switch through a reversible disulfide bond that is formed between two catalytic cysteine residues (Cys88 and Cys378)[18]. As it might be difficult to further overexpress ThlA and as its biochemical properties might also contribute to the low C4/C2 ratio in CAB1058 (Fig. 2b), ThlA was replaced by AtoB, a thiolase from *E. coli* that has a higher catalytic efficiency, is less sensitive to CoA-SH[17,19,20] and not subject to a redox-switch. For this purpose, a synthetic *atoB* gene was designed, codon harmonized[21,22] for *C. acetobutylicum*, and used to replace in frame *thlA* in CAB1058 (to yield CAB1059). In CAB1059, the butanol to ethanol ratio (mol/mol) and the butanol flux (in % of the glucose flux) only increased from 0.59 to 0.84 (Fig. 2b) and from 45 to 55.1, respectively, while the thiolase activity increased 1.7 fold (Supplementary Fig. 2) suggesting that the C4/C2 ratio is not controlled by the thiolase level or by the levels of the other enzymes of the pathway and might be thermodynamically controlled. The condensation of two acetyl-CoA to acetoacetyl-CoA catalyzed by the thiolase is thermodynamically unfavorable[23]. Therefore, it is important that acetoacetyl-CoA is efficiently reduced by Hbd (Fig. 1b) to pull the reaction[23]. Hbd is an NADH-dependent enzyme[24] and, as it likely works near the thermodynamic

**Table 1 Representative studies of high level butanol production in *C. acetobutylicum* and *C. beijerinckii***

| Strain | Type | Substrate/Medium | Fermentation Type | Butanol Yield (g/g) | Butanol Titer (g/L) | Butanol Productivity (g/L/hr) | Reference |
|---|---|---|---|---|---|---|---|
| *C.acetobutylicum* ATCC824 | Wild type | Glucose/synthetic medium | Batch | 0.18 | 10 | 0.25 | Ref.[38] |
| *C.acetobutylicum* ATCC824 | Wild type | Glucose/synthetic medium + 1 mM MV | Batch | 0.27 | 13.5 | 0.22 | Ref.[38] |
| *C.acetobutylicum* ATCC824 | Wild type | Glucose/synthetic medium | Continuous chemostat | 0.185 | 8.8 | 1.1 | Ref.[33] |
| *C.acetobutylicum* XY16 | Wild type | Glucose/complex medium | Continuous immobilized | 0.31 | 5.7 | 11.3 | Ref.[39] |
| *C.acetobutylicum* ATCC824 | Wild type | Glucose/complex medium | Continuous extractive pervaporation | 0.23 | 132 | 0.74 | Ref.[40] |
| *C.acetobutylicum* ATCC824 (pGROE1) | Engineered | Glucose/complex medium | Batch | NA | 17.1 | 0.14 | Ref.[41] |
| *C.acetobutylicum* HKKO | Engineered | Glucose/complex medium | Batch | 0.2 | 18.2 | 0.38 | Ref.[42] |
| *C.acetobutylicum* BEKW (pPthlAAD) | Engineered | Glucose/complex medium | Batch | 0.29 | 18.9 | 0.33 | Ref.[43] |
| *C.acetobutylicum* ATCC824 *pfkA + pykA* | Engineered | Glucose/complex medium | Fed-batch | 0.21 | 19.1 | 0.19 | Ref.[44] |
| *C.acetobutylicum* BKM19 | Engineered | Glucose/complex medium | Continuous cell recycle | 0.17 | 11.9 | 10.7 | Ref.[45] |
| *C.acetobutylicum* CAB1060 | Engineered | Glucose/synthetic medium | Continuous extractive distillation | 0.35 | 550 | 14 | This study |
| *C. beijerinckii* P260 | Wild type | Glucose/complex medium | Batch | 0.26 | 11.8 | 0.25 | Ref.[46] |
| *C. beijerinckii* P260 | Wild type | Glucose/complex medium | Batch vacuum fermentation | 0.20 | 35.9 | 0.27 | Ref.[46] |
| *C. beijerinckii* DSM 2152 | Wild type | Glucose/complex medium | Continuous extractive gas striping | 0.19 | NA | 0.63 | Ref.[47] |

equilibrium, the acetoacetyl-CoA/3-hydroxybutyryl-CoA ratio will be dependent on the NADH/NAD$^+$ ratio. As it has been demonstrated in solventogenic *C. acetobutylicum* cells that the NADPH/NADP$^+$ ratio is at least 70 times higher than the NADH/NAD$^+$ ratio[25], we anticipate that it would be advantageous to replace Hbd by a strictly NADPH-dependent enzyme to decrease the acetoacetyl-CoA/3-hydroxybutyryl-CoA ratio and potentially improve the butanol/ethanol ratio. Such an enzyme has been identified and characterized in *Clostridium kluyveri*[26]. In strain CAB1060, we have replaced *hbd* by *hbd1* from *C. kluyveri*. A drastic improvement in the butanol/ethanol ratio (Fig. 2b) and the *n*-butanol fluxes (Supplementary Fig. 1) were observed with, respectively, 6- and 1.6-fold increase. Enzyme assays on crude extract confirm that in the genetic modification of CAB1060, the NADH-dependent 3-hydroxybutyryl-CoA dehydrogenase activity was very low while the NADPH-dependent activity was high; the reverse was observed for CAB1059 (Supplementary Fig. 2). CAB1060 produces *n*-butanol at high yield (0.34 g/g) (Table 1) in a simple mineral media without the addition of an organic nitrogen source. Attempt to abolish acetate production in the CAB1060 strain by either deleting *pta* and *ack* (that code for the phospho-transacetylase and the acetate kinase involved in the last two steps of acetate formation) or *hydA* that codes for the main hydrogenase of *C. acetobutylicum* were unsuccessful suggesting that such mutants were not viable. The *pta-ack* mutant might not be viable due its inability to redirect all the electron flow from hydrogen production to NADH production using the Ferredoxin NAD$^+$ reductase enzyme while the *hydA* mutant might not be viable due to its inability to either avoid acetate production or to re-oxidize reduced ferredoxin to produce NADH or both. If a method to simultaneously inactivate several genes would be available, it would have been interesting to delete at the same time *pta*, *ack*, and *hydA* as such a strain might be viable: eliminating both acetate and hydrogen would lead to a strain converting glucose to butanol and ethanol with a perfectly equilibrated redox balance.

The stability of CAB1060 was evaluated in chemostat culture. We could maintain stable *n*-butanol (between 9 and 10 g/L) and biomass concentrations (Fig. 3a) for more than 45 days and after 13 days of cultures only slight changes in the ethanol/acetate ratio

between 1 and 0.8 (mol/mol) were observed (Fig. 3a and Supplementary Fig. 1).

**Development of continuous process for *n*-butanol production**. Having a stable strain that produces *n*-butanol at a high yield in a continuous culture is a prerequisite, but it is clearly not sufficient for a commercial process as both the titer and the productivities are too low. To improve those two parameters, several methods of in situ butanol recovery including gas stripping, vacuum fermentation, pervaporation, liquid–liquid extraction, perstraction, and adsorption, have been investigated[27]. However, high volumetric productivities or scaled-up industrial production are still hard to achieve. We then envisioned a continuous high cell density bioreactor with in situ extraction of the alcohols (*n*-butanol + ethanol) by distillation under low pressure (0.18–0.46 × 10$^4$ Pa, for a boiler temperature regulated between 35 and 37 °C) without using any microfiltration membrane (Fig. 4). The advantages of this method are (1) that the fermentor is at atmospheric pressure which allows the fermentation gas to be eliminated without pumping and limit the cost for low pressure distillation compared to vacuum fermentation and (2) high cell density can be obtained without the use of microfiltration membranes. Three runs of continuous extractive fermentation were performed (Supplementary Table 1) but only the run III will be presented in detail below. After an initial batch phase without distillation, this setup was run in a fed-batch mode by (i) starting vacuum distillation and (ii) exponentially feeding the reactor with a medium at a high concentration of glucose (~530 g/L) while maintaining full cell retention until reaching an OD 600 nm of 90–100 (Fig. 3b), which corresponds to ~28–30 g/L cell dry weight. As shown in Fig. 3b, the targeted cell density was reached in ~70 hours after vacuum distillation and feeding began, at which point a cell bleeding was implemented at a dilution rate of 0.05 h$^{-1}$ to maintain a constant cell density and perform the fermentation in a continuous mode. The concentration of *n*-butanol in the fermentor was between 9 and 10.5 g/L (Supplementary Fig. 3) while the average concentration of *n*-butanol in the butanol-rich phase that was continuously collected was 550 g/L with concurrent average

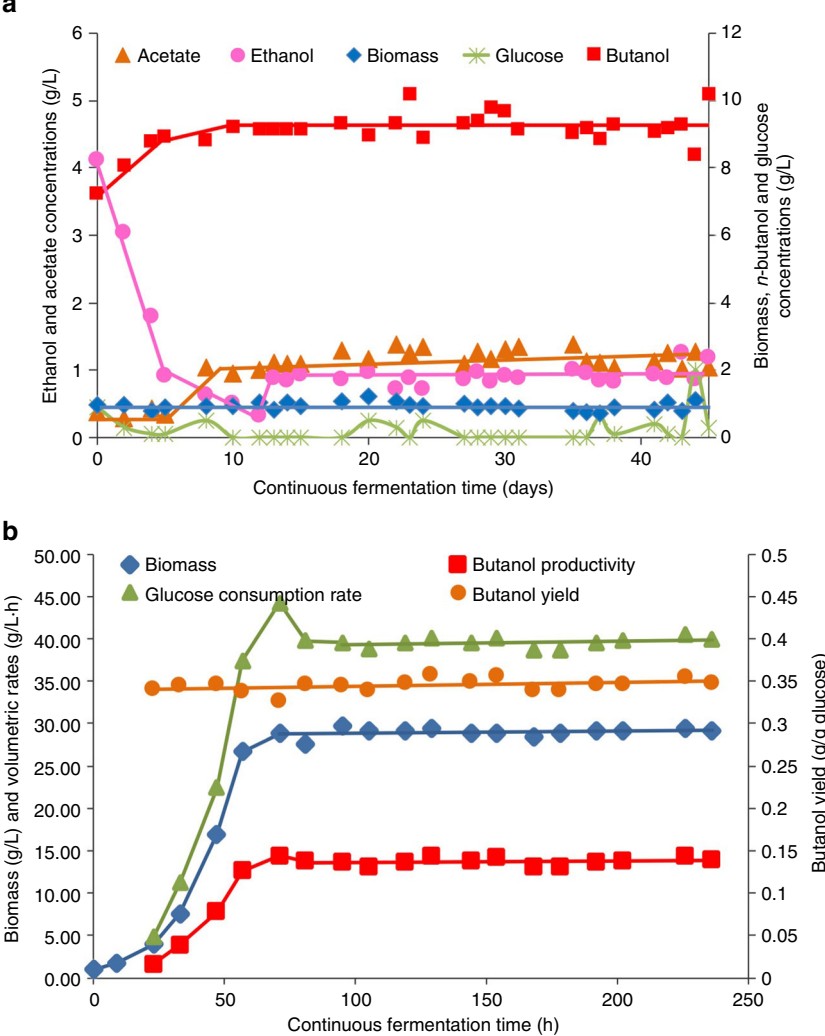

**Fig. 3** Continuous cultures of *C. acetobutylicum* CAB1060. **a** In phosphate-limited chemostat culture at pH 5, **b** in phosphate-limited high cell density culture at pH 5 with in situ extraction of the alcohols (*n*-butanol + ethanol) by distillation under low pressure. The arrow indicate the start of the cell bleeding. For both continuous cultures, the initial batch phase is not presented

ethanol and acetate titers of 60 and 3.3 g/L (Supplementary Fig. 3). Starting at hour 70 and for 170 hours afterwards (i.e., up to 240 hours of fermentation time), the volumetric productivities of butanol and ethanol remained essentially constant, along with the glucose volumetric consumption. For the three runs (Supplementary Table 1), the average volumetric productivities of *n*-butanol and ethanol were 14 and 1.5 g/L/hr, respectively, and the average glucose volumetric consumption rate was 39.4 g/L/hr. This resulted in an average *n*-butanol yield of 0.35 g/g (~84% of the theoretical maximum) and *n*-butanol + ethanol yield of 0.39 g/g[28]. Such high values of *n*-butanol productivities, titer, and yield compare very well with the performances of the best corn wet milling continuous industrial ethanol processes that display ethanol titers of ~100 g/L, yield of 0.46 g/g (~90% of the theoretical maximum) and productivity of 10 g/L/hr[29]. In the US the selling price of *n*-butanol is 2.3 times higher than ethanol ($1.2 per kg versus $0.5 per kg, ISIS data of September 2017) while the yield of alcohol formation of our process is only 1.25 lower than the ethanol process. If scaled-up at the industrial scale, the technology presented here should drastically decrease both the operational and capital expenditure costs of the historical Weizmann process and make the commercial production of *n*-butanol economical and more attractive than the production of ethanol.

## Methods

**Bacterial strains, plasmids, and primers**. The bacterial strain and plasmids used in this study are listed in Supplementary Table 2 and Supplementary Table 3. The specific oligonucleotides used for PCR amplification were synthesized by Euro-gentec (Supplementary Table 4).

**General protocol for gene deletion**. Gene disruption in specified chromosomal loci was carried out by homologous recombination[13] with the modifications of patent application WO2008/040387[30]. Deletion cassettes consisting of the upstream homologous region-FRT-catP-FRT-downstream homologous region were cloned in the pSOS95-MLSr-upp plasmid.

The pSOS95-MLSr-upp plasmids with the different deletion cassettes were used to transform the successive recipient *C. acetobutylicum* strains which were constructed from the MGCΔ*cac1502*Δ*upp* platform strain[13].

As a general procedure, thiamphenicol-resistant transformants were selected on CGM 50 g/L Glucose 0.1 M MES pH 6.1 (CGMMG) plates containing thiamphenicol (10 μg/mL) (Tm). One colony was cultured for 24 hours in liquid CGMMG with thiamphenicol (10 μg/mL) and 100 μL of undiluted culture was plated on CGMMG with thiamphenicol (10 μg/mL) and 5-FU (1 mM). Colonies resistant to both thiamphenicol and 5-FU were replica plated on both CGMMG with thiamphenicol (5 μg/mL) and CGMMG with clarithromycin (40 μg/mL) to select clones where 5-FU resistance is also associated with clarithromycin sensitivity. The genotype of the clones resistant to thiamphenicol and sensitive to clarithromycin was checked by PCR.

The *catP* resistance cassette was removed by transforming the strain with plasmid pCLF1 expressing the flp1 gene encoding the Flp recombinase from *S. cerevisiae*[13]. One colony of a clarithromycin resistant clone was cultured on

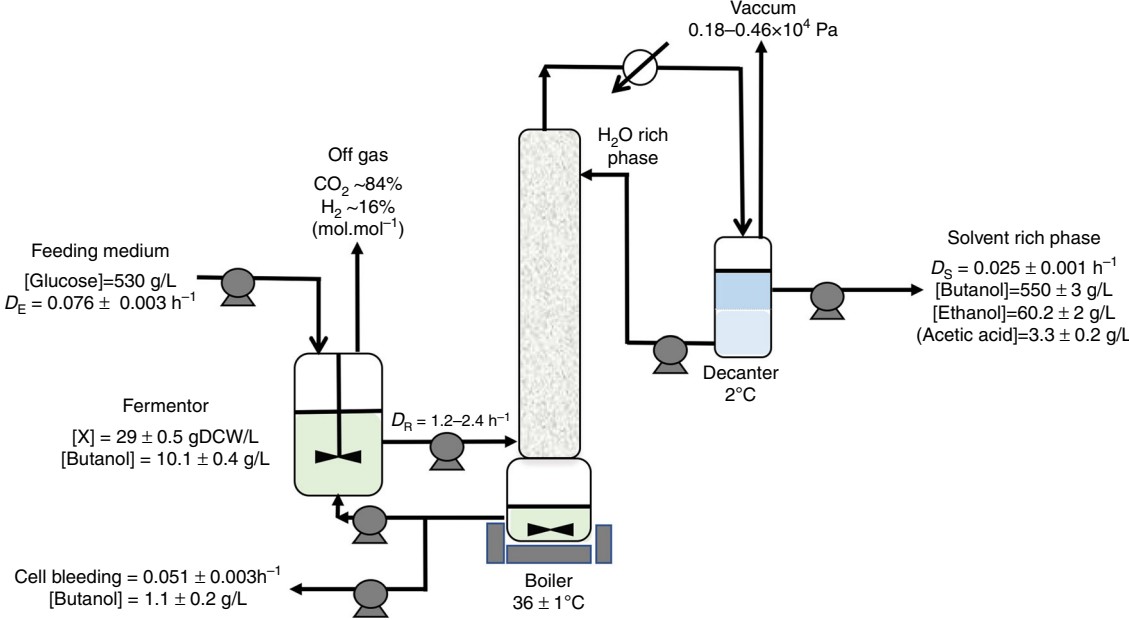

**Fig. 4** Schematic representation of the extractive continuous fermentation process. The fermentor, boiler, and decantor have working volumes of 0.5 L, 0.3 L, and 0.1 L, respectively. For more details see Methods

liquid CGMMG without clarithromycin and the culture was plated on CGMMG with 5-FU (1 mM). Colonies resistant to 5-FU were replica plated on CGMMG with clarithromycin (40 μg/mL), CGMMG with thiamphenicol (10 μg/mL), and CGMMG without antibiotic to select clones. The 5-FU resistance was associated with the loss of the pCLF1 vector and the thiamphenicol sensitivity to the elimination of the *catP* marker. The genotype of sensitive clones was checked by PCR.

CAB1057 was constructed from the MGCΔcac1502Δupp platform strain by successively applying this protocol to the deletion of *ptb-buk, ctfAB*, and *ldhA*[12,13,30]. CAB1058 was constructed from CAB1057 by applying the same protocol to the deletion of *CA_C2713* encoding the Rex regulator and the removal of the *catP* resistance cassette.

**Construction of pEryuppatoB plasmid and CAB1059 strain**. A synthetic *atoB* operon coding for the AtoB thiolase of *E. coli* was synthesized as follow. A synthetic *atoB* gene was designed with both an optimized synthetic RBS (translation initiation rate was calculated to be 60217, a value similar to the translation initiation rate of *thlA*) and a harmonized codon usage for *C. acetobutylicum*. The synthetic gene was synthesized by Life technology (*atoB*s) with a *Bam*HI site in 5′ and a *Sfo*I site in 3′ and was cloned in a *Bam*HI-*Sfo*I digested pSOS95 to yield pSOS95atoBs[31].

The synthetic *atoB* gene was PCR amplified with the Phusion DNA polymerase with pSOSatoBs as a template and the Atob-1 and Atob-2 oligonucleotides as primers. This DNA fragment was directly cloned into the *Stu*I linearized pEryUppAthlA (for the construction of pEryUppAthlA[22]) by homologous recombination using the GeneArt® Seamless Cloning & Assembly kit to yield the pEryUppAtoB plasmid.

The pEryUppAtoB plasmid was used to transform by electroporation to CAB1058 strain. After selection on Petri plates for clones having inserted the pEryUppAtoB plasmid by homologous recombination (resistant to erythromycin 20 μg/mL), two colonies were cultured for 24 hours in CGM Glucose MES 0.1 M medium. Appropriate dilutions were plated on CGM Glucose MES 0.1 M (CGMMG) with 5-FU 1mM. To select integrants having excised and lost pEryUppAtoB, 5-FU resistant clones were replica plated on both CGMMG with 5FU and CGMMG with erythromycin at 40 μg/mL. To identify clones having lost pEryUppAtoB and possessing an *atoB* insertion, (clones resistant to 5-FU and sensitive to erythromycin) were checked by PCR analysis (with primers Thl-0 and Thl-5 located outside of the AthlA locus and Atob-1 and Atob-2 primers binding specifically to *atoB* gene). The clones that had *thlA* replaced by *atoB* were isolated to obtain pure clones to obtain CAB1059 strain.

**Constructing pSOS95-upp-hbd1-catP-oriRepA and CAB1060 strains**. To amplify the two homologous regions flanking the CDS region of the *hbd* gene on the genome of *C. acetobutylicum* ATCC824, two couples of primers (primer 1-hbd1-sacII/primer 2 and primer 5/primers 6-hbd1-sacII) were used. Two fragments of 909 bp and 989 bp were obtained. Primer 3 and primer 4 were used to amplify the *hbd1* gene from genomic DNA of *C. kluyveri* DSM555 to obtain the third fragment of 886 bp. Fragment 1 and 3 was fused by overlapping PCR using

primer 1-hbd1-sacII and primer 4 to obtain the fourth fragment of 1778 bp. After that, this fourth fragment was fused with the second fragment by primer 1-hbd1-sacII and primer 6-hbd1-sacII to produce the fifth fragment of 2692 bp.

The *colE1* origin on pSOS95-MLSr-upp[30] was replaced by the *repA* origin of pSC101. The *repA* origin was amplified from pSC101 by two primers oriRepA-XbaI-F and oriRepA-5'R to obtain a PCR product of 2296 bp. The partial ampicillin resistant gene from pSOS95-upp was amplified by Amp3'-F and Amp-ScaI-R to obtain a PCR product of 573 bp. The two resultant PCR fragment was then fused by PCR using oriRepA-XbaI-F and Amp-ScaI-R. This 2849 bp fragment was introduced to pSOS95-upp to replace *repL* fragment by digestion with *Xba*I and *Sca*I restriction enzymes and ligation to obtain pSOS95-upp-orirepA. After that, the homologous fragment containing *hbd1* CDS was introduced to this pSOS95-upp-orirepA by the *Sac*II site to obtain pSOS95-hbd1-orirepA plasmid. The *catP* cassette (flanking by FRT sequences) was inserted into the *Stu*I site carried by primer 4 and primer 5 in the homologous region to generate the final plasmid pSOS95-hbd1-catP-oriRepA.

The final plasmid pSOS95-hbd1-catP-oriRepA was used to transform into CAB1059 strain and followed the protocol presented above for the deletion[30]. The transformants on plates CGMMG Thiamphenicol 10 μg/mL were selected and checked by PCR for the single integration by two couples of primers etfB-3-F and catP-3D or cac2706-3Rb and catP-5R. The integrants were spread on plates CGMMG containing Thiamphenicol 10 μg/mL and 5-FU 1 mM to promote the second integration. Clones that were Thiamphenicol resistant and Clarithromycin sensitive were selected and checked for double cross-over events by PCR using two external primers etfB-3-F and cac2706-3Rb. The mutants were isolated to obtain CAB1060-catP strain. After that, pCLF1 was introduced into CAB1060-catP strain to remove *catP* cassette to generate the marker-less strain CAB1060.

**Chemostat culture of recombinant strains**. The strains were evaluated in phosphate-limited continuous culture[32] fed at 30 g/L glucose at a pH of 5.0. Metabolic flux analysis was performed using the genome scale model previously developed[16].

**Continuous extractive high density cell recycle fermentation**. A 10% inoculum of mid-exponential phase (OD600nm of 0.8–1.5) CAB1060 culture was used to inoculate a 1 L glass bioreactor with a 0.5 L working volume of anaerobic synthetic medium[33] with 60 g/L glucose. Fermentation pH was controlled from dropping below 5.0 using 4 M NH4OH. Once the cultures reached an OD600nm of three, extractive fermentation using low pressure distillation was initiated. Low pressure distillation was accomplished by sending the whole broth to a 3 L Pyrex boiler (0.3 L working volume)[34] equipped to maintained a constant temperature (35–37 °C) and a constant low pressure (0.18–0.46 × 10⁴ Pa) above the boiler. The distillation column has an internal diameter of 7.8 cm, a height of 60 cm and was filed with mesh column packing from Multiknit Ltd. The decanter was maintained at 2 °C, has a working volume of 0.1 L, and consists of a Pyrex tube with an internal diameter of 2.4 cm and a height of 25 cm. The initial residence time of the broth in the boiler was 0.5 hour. The fermenter was fed exponentially with a feed medium (see the composition below) containing high glucose concentration (530 g/L) by

just keeping the volume of the fermentor constant. The distillate was condensed (2 °C) and de-mixed in a butanol-rich- and a water-rich phase; the water-rich phase was recycled to the distillation column while the butanol-rich phase was collected. Once cell concentration reached an $OD_{600nm}$ of 90–100 (28–30 g/L cell dry weight), a cell bleeding was started at a dilution rate of 0.05 h$^{-1}$ while still keeping volume of the fermenter constant by the addition of feed medium. The residence time of the broth in the boiler was decreased to 0.25 h.

The feeding medium has the following composition (per liter of tap water): glucose, 530 g; $KH_2PO_4$, 1.8 g; KCl, 5.7 g; $MgSO_4.7H_2O$, 3.6 g; $FeSO_4.7H_2O$, 0.5 g; $NH_4Cl$, 18 g; Biotin, 0.18 mg; and PAB, 0.14 g.

**Measurement of fermentation parameters**. Biomass concentration was determined by a DCW method that includes three steps: a centrifugation (16,000 g, 5 min, room temperature) of 1.5 mol broth in an Ependorf tube, two washes with Milli-Q water, and drying under vacuum at 80 °C. The concentrations of glucose, and of all the fermentation products were determined based on high-performance liquid chromatography (HPLC)[35] using a HPX87H (Biorad) column, H2SO4 at 0.5 mM, as mobile phase and a refractometer to quantify the different compounds.

**Enzyme activity measurements**. Cell crude extract of *C. acetobutylicum* mutants was prepared in 100 mM Tris-HCl (pH 7.6), 5 mM dithiothreitol, and 2.5% glycerol. The cell suspension was sonicated in an ultrasonic disintegrator (Vibracell 71434; Bioblock, Illkirch, France), at 0 °C in four cycles of 30 s with 2-min intervals between each cycle. Cell debris was removed by three centrifugations at 13,000× *g* for 5 min. The supernatant was loaded to PD10 column.

Thiolase activity was measured in the reverse thiolytic cleavage direction. The reaction containing 0.1 M Tris-HCl pH8; 0.2 mM CoASH; 60 μM Acetoacetyl-CoA; 10 mM $MgCl_2$; and 1 mM DTT and diluted crude extracts was incubated at 30 C and followed the decrease of acetoacetyl-CoA at 303nm[23,36]. The enzyme activity was calculated by the difference in the slope values of the sample and control and using the molar extinction coefficient of 14,000 M$^{-1}$ cm$^{-1}$.

HBD activity was measured in the physiological direction[36,37]. The reaction containing 0.1 mM MOPS (pH 7.0) buffer, 1 mM DTT, 0.1 mM acetoacetyl-CoA, and 0.15 mM NADH, and crude extract at different dilution was incubated at 35 °C. The decreasing of NADH was followed spectrophotometrically at 345 nm. The enzyme activity was calculated by the difference in the slope values of the sample and control and using the molar extinction coefficient of 6220 M$^{-1}$ cm$^{-1}$.

HBD1 activity was measured in the physiological direction[26]. Reaction components were 0.1 M potassium phosphate, 25 mM potassium citrate, 0.15 mM NADPH, 0.6 mM acetoacetyl-CoA, and crude extract. The assay was done at 35 °C and the decrease of NADPH was followed at 340 nm. The molar extinction coefficient of 6220 M$^{-1}$ cm$^{-1}$ was used to calculate the enzyme activity.

## Data availability

All relevant data are included with the manuscript (as figure or supplementary information files). All data are available upon reasonable request from the corresponding author.

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

## Acknowledgements

This work was financially supported by the European Community's Seventh Framework Program "CLOSTNET" (PEOPLE-ITN-2008-237942) (to TN) and by Metabolic Explorer Company.

## Author contributions

N.P.T.M and C.R. designed and performed experiments. I.M.S. and P.S. conceived this study, analyzed the data, discussed results, and wrote the manuscript.

## Additional information

**Competing interests:** Three patent applications have been filed related to this work. C.R. and P.S. hold shares in Metabolic Explorer, a company pursuing applications of part of these technologies. The remaining authors declare no competing interests.

