## [Peer Review File · Nature Communications]

Reviewers' comments:

Reviewer #1 (Remarks to the Author):

This manuscript presents an interesting metabolic and process engineering approaches for enhanced butanol production in ABE fermentation by *Clostridium acetobutylicum*. Through step-by-step and rational metabolic pathway modification, mutant CAB1060 produced n-butanol at a very high yield (34 wt%). Especially, replacing Hbd by a strictly NADPH dependent enzyme showed an amazing result that decreased the acetoacetyl-CoA/3hydroxybutyryl-CoA ratio and dramatically improved the butanol/ethanol ratio. In addition, through continuous high cell density bioreactor with in situ extraction, the final titer and productivity present a great potential for the commercial production of n-butanol.

Overall, this work would have a good potential for industrial application if the vacuum distillation can be scaled up. The authors exaggerated by claiming that "n-butanol at high yield (34 wt%) has never been achieved with any microorganism" and "Such high values of n-butanol productivities, titer, and yield have never been obtained.." There have been several studies reporting n-butanol yield of greater than 0.34 g/g, productivity of greater than 14 g/L.h, and titer of greater than 550 g/L, although these were achieved separately in different studies instead of all in one process. The authors thereby should acknowledge this fact and clarify that the accomplishment is to achieve all these performance metrics in the same process.

Several technical issues require further discussion:

1. Why hydrogen production was significantly reduced in CAB1058, 1059 and 1060? The metabolic engineering used in these studies did not appear to have a direct effect on hydrogen production (perhaps indirectly through their effects on NADH balance, but how?)
2. Why $(C_4+C_3)/C_2$ was still lower for CAB1060 compared to CA824? Same question for Butanol/Ethanol? One would expect increased C_4/C_2 ratio after increasing flux from acetyl-CoA to Butyryl-CoA with the Hbd1.
3. What about acids (acetate and butyrate) production? Also, was there any acetone production after ctfAB knockout? Figure 2 should include data on acids and acetone for various strains.
4. How were CO₂ and H₂ measured during the fermentation? There is no method given.
5. Fig 3 - the concentrations of glucose, acetate, butyrate, ethanol, and acetone (if present) should also be included so one can get the full set of fermentation kinetic data. Fig. 3B should also present the glucose and butanol concentrations in the fermentor (and boiler if different) and decanter. It is not clear how the butanol productivity and yield were calculated without seeing the kinetic data. Also, what was the glucose conversion rate or residual glucose?
6. Need more details about the set-up shown in Fig. 4. What was the recirculation rate between the fermentor and boiler? It said the residence time in the boiler was between 0.25 and 0.5 h (how this

was varied with what?) How was the exponential feeding was done? does the arrow indicating cell bleeding in Fig 3B the time the continuous feeding started?

7. I suppose the mutant was stable in the process without using any antibiotics. What about the "slight changes in ethanol/acetate ratio" observed (cannot be seen in Fig, S1)? This should be more specific and why there would be changes in the ratio in the chemostat culture?

8. The continuous culture with vacuum distillation started when in Fig. 3B? Table S1 indicates the fed-batch was 69 h. Was this time included in the 250 h shown in Fig. 3B? A better and more detailed figure caption should be given for Fig. 3.

9. A comparison table listing butanol yield, titer, and productivity achieved in previous studies by others should be given (may be as a supplementary table). Additional references about these notable studies should also be given.

10. Why vacuum distillation? Any advantages over other in situ butanol separation methods? This should be briefly discussed, especially to address the concern on its scalability.

11. After replacing Hbd by a strictly NADPH dependent enzyme, why Thl activity also increased substantially? ThIA did not knock-out?

12. As shown in Fig. S1, although butanol to ethanol ratio was increased to 0.84, CAB 1059 model present a lower alcohol flux with the increment of acetate production as compared to the CAB1058 model. Whether ThIA replaced by AtoB is a good approach for butanol production?

After properly addressing the above issues, I think the work is appropriate to be published in the journal.

Reviewer #2 (Remarks to the Author):

In this ms, Soucaille and collaborators carried out an extensive metabolic engineering of *C. acetobutylicum* to generate as strain that enables superior butanol/solvent titers and yields. They then integrated this strain into a continuous process that engages vacuum distillation to achieve an exceptional performance that exceeds all prior art and has all the characteristics of an economically attractive industrial process. The metabolic engineering work is tremendous, in its vision, design and execution, and exceeds all prior state of the art.

Overall, this is a superb achievement and story, and I enthusiastically recommend publication. Having said that, this current version is a very short format that one would suspect was submitted for publication to Nature or Nature Biotechnology. As such, the story telling is too dense for most readers, especially for the non-specialist readers. I would therefore recommend that they recast the paper into the longer Nature Communications format, especially the metabolic engineering work.

They can bring into the main ms a good deal of the now supplemental material.

Reviewer #3 (Remarks to the Author):

The manuscript by Nguyen et al. describes an optimized process for microbial n-butanol production from glucose. The authors describe a rational approach for metabolic engineering of *Clostridium acetobutylicum* to minimize by-product formation and maximise the yield of n-butanol.

Furthermore, they established a continuous production process employing this strain, which combines a higher cell density than usual with in-situ extraction of the toxic product by distillation.

The obtained yield and productivity exceed values published up to now and could lead to a new economical exploitation of this historic process.

The details given appear appropriate, and reproducibility (incl. statistics) has been sufficiently shown in this reviewer's opinion. However, the manuscript itself is quite unpleasant to read as all genes are named with their abbreviations and most of them are not spelled out. It is recommendable to use the correct names of all enzymes – it might seem lengthy but it makes the text much more readable, particularly for readers who are not entirely in this field. It is also not appropriate to name microorganisms just with a number and not with their correct name. The English should be checked once more, there are some errors, such as “might be difficult to further overexpressed” or “its biochemical properties might also contributes” – there are more, which I do not list here.

This reviewer disagrees with the simplified notion that the decline of the historic fermentation process was solely due to the development of petrochemistry. Clearly, petrochemistry played a significant role. However, also a drastic increase of the substrate costs was decisive. (Molasses were newly used for feed, which was much more profitable.) Other factors were involved as outlined in detail by Jones&Wood, who were cited by the authors. This factor appears important, because it will gain significance also nowadays, when more and more microbial processes start to compete for sugar.

The last phrase notes that the new process would be economical. Could the authors give an estimate in numbers here? Why do they think so? Based on current glucose costs, which would be feasible n-butanol costs with their process? Ethanol for example is mostly not economical (outside of Brazil), when subsidies are taken out of the system, so this comparison makes it difficult to judge what the authors mean.

Can the authors compare their technology to Gevo's technology for isobutanol production with yeasts? The Gevo technology is also based on in-situ removal of the isobutanol – the organisms is clearly completely different. This comment is not meant to lengthen the manuscript unduly – it would be just interesting to have a comparison to a (semi) commercial process, which is more closely related than ethanol production.

Michael Sauer, BOKU Vienna

Reviewers' comments:

Reviewer #1 (Remarks to the Author):

This manuscript presents an interesting metabolic and process engineering approaches for enhanced butanol production in ABE fermentation by Clostridium acetobutylicum. Through step-by-step and rational metabolic pathway modification, mutant CAB1060 produced n-butanol at a very high yield (34 wt%). Especially, replacing Hbd by a strictly NADPH dependent enzyme showed an amazing result that decreased the acetoacetyl-CoA/3hydroxybutyryl-CoA ratio and dramatically improved the butanol/ethanol ratio. In addition, through continuous high cell density bioreactor with in situ extraction, the final titer and productivity present a great potential for the commercial production of n-butanol.

Overall, this work would have a good potential for industrial application if the vacuum distillation can be scaled up. The authors exaggerated by claiming that "n-butanol at high yield (34 wt%) has never been achieved with any microorganism" and "Such high values of n-butanol productivities, titer, and yield have never been obtained.." There have been several studies reporting n-butanol yield of greater than 0.34 g/g, productivity of greater than 14 g/L.h, and titer of greater than 550 g/L, although these were achieved separately in different studies instead of all in one process. The authors thereby should acknowledge this fact and clarify that the accomplishment is to achieve all these performance metrics in the same process.

Response to reviewer 1: We agree with reviewer 1 that high n-butanol yield of 34% has been obtained in other organisms but it was in complex media containing high amount of organic sources of nitrogen (yeast extract and/or tryptone in most cases), never on a simple mineral media with only one source of carbon and energy like the one use in this study. This information has been added in the manuscript. Reviewer 1 is also right in the fact that separately high titers and productivities have been obtained but the three parameters have never been obtained together. This has been clarified in the revised manuscript

Several technical issues require further discussion:

1. Why hydrogen production was significantly reduced in CAB1058, 1059 and 1060? The metabolic engineering used in these studies did not appear to have a direct effect on hydrogen production (perhaps indirectly through their effects on NADH balance, but how?)

Response to reviewer 1: the decrease in hydrogen production has been obtained by redirecting the carbon fluxes in reduced compounds (butanol and ethanol) at the expense of more oxidized compounds like acetone or butyrate and then increasing the NADH demand. Attempt to redirect the electron fluxes by knocking out the hydrogenase encoding genes has been impossible even in the CAB1060 strain: such a mutant strain appears not to be viable. Possible explanations would be that either the level of ferredoxin-NAD⁺ reductase is too low but has the gene responsible for this activity is unknown this hypothesis could not be tested or that acetate production cannot be sufficiently decrease to equilibrate the redox balance in n-butanol and ethanol production. The manuscript has been modified to present these hypotheses.

2. Why (C4+C3)/C2 was still lower for CAB1060 compared to CA824? Same question for Butanol/Ethanol? One would expect increased C4/C2 ratio after increasing flux from acetyl-CoA to Butyryl-CoA with the Hbd1.

Response to reviewer 1: This is due to the fact that the wild type strain consumes acetate and acetoacetyl-CoA to produce acetyl-CoA and acetone. When this pathway is deleted the strain accumulate acetate and acetoacetyl-CoA. Regarding the butanol to ethanol ratio, we know that AdhE1 replaces AdhE2 for ethanol and butanol production in the CAB1060 strain. Km for acetyl-CoA and butyryl-CoA are probably different for each enzyme and might explain the differences.

3. What about acids (acetate and butyrate) production? Also, was there any acetone production after ctfAB knockout? Figure 2 should include data on acids and acetone for various strains.

Response to reviewer 1: All the engineered strain produced no acetone, no butyrate and no lactate. This can be seen from the flux analysis presented in Fig S1 in supplementary material. Data on acetate production have been added in figure 2 and figure 3.

4. How were CO2 and H2 measured during the fermentation? There is no method given.

Response to reviewer 1: Information on H₂ and CO₂ production measurement has been added in the revised manuscript.

5. Fig 3A - the concentrations of glucose, acetate, butyrate, ethanol, and acetone (if present) should also be included so one can get the full set of fermentation kinetic data. Fig. 3B should also present the glucose and butanol concentrations in the fermentor (and boiler if different) and decanter. It is not clear how the butanol productivity and yield were calculated without seeing the kinetic data. Also, what was the glucose conversion rate or residual glucose?

Response to reviewer 1: As stated above, there was no butyrate, no lactate and acetone produced. Concentration of acetate, ethanol and glucose have been added in fig 3A. It would make the figure 3B too complex if the concentrations of glucose, ethanol and n-butanol in the fermentor, in the boiler and in the decanter, were added in Figure 3B. We have then added a new figure, Fig. S3 with these concentrations in supplementary

material.

6. *Need more details about the set-up shown in Fig. 4. What was the recirculation rate between the fermentor and boiler? It said the residence time in the boiler was between 0.25 and 0.5 h (how this was varied with what?) How was the exponential feeding was done? does the arrow indicating cell bleeding in Fig 3B the time the continuous feeding started?*

Response to reviewer 1: The recirculation rate was increased from 0.6 to 1.2 l/h when the cell bleeding was started. These flow rates were determined from previous experiments not presented here. Regarding the exponential feeding, we just kept the volume of the fermentor constant and compensate the volume of the distillate by a feeding medium with high glucose concentration. When cell bleeding was started, the strategy was the same: feeding rate was based on the maintenance of a constant volume in the bioreactor and was the sum of the bleeding rate and the rate of distillate going out the bioreactor. These informations have been added in the material and method.

7. *I suppose the mutant was stable in the process without using any antibiotics. What about the "slight changes in ethanol/acetate ratio" observed (cannot be seen in Fig, S1)? This should be more specific and why there would be changes in the ratio in the chemostat culture?*

Response to reviewer 1: There is no plasmid (except the megaplasmid) or antibiotic marker in the CAB1060 strain. We observed a change in the ethanol/acetate ratio (mol/mol) from 1 to 0.8. This information was added in the manuscript. We have no explanation for this slight change.

8. *The continuous culture with vacuum distillation started when in Fig. 3B? Table S1 indicates the fed-batch was 69 h. Was this time included in the 250 h shown in Fig. 3B? A better and more detailed figure caption should be given for Fig. 3.*

Response to reviewer 1: The only part of the fermentation that is not presented is the batch phase (As was explained in supplementary material). The vacuum distillation was then started at t=0 in Fig.3B and the 250h include the fed-batch. We try to make it clear in the new figure caption and the description in material and methods.

9. *A comparison table listing butanol yield, titer, and productivity achieved in previous studies by others should be given (may be as a supplementary table). Additional references about these notable studies should also be given.*

Response to reviewer 1: A comparison table of other studies using solventogenic clostridia as well as references have been added.

10. *Why vacuum distillation? Any advantages over other in situ butanol separation methods? This should be briefly discussed, especially to address the concern on its scalability.*

Response to reviewer 1: Without cell separation, vacuum distillation allows higher extraction rate than gas stripping or pervaporation. A small discussion was added in the revised manuscript.

11. *After replacing Hbd by a strictly NADPH dependent enzyme, why Thl activity also increased substantially? ThIA did not knock-out?*

Response to reviewer 1: In the CAB1060 strain *thIA* is replaced by a synthetic *atoB* and is then not present. One explanation might be that *thIB* presents on the pSOL1 megaplasmid which is normally not expressed might be now expressed under those new conditions. As nothing is known regarding *thIB* regulation by ThIR, it is difficult to speculate on this.

12. *As shown in Fig. S1, although butanol to ethanol ratio was increased to 0.84, CAB 1059 model present a lower alcohol flux with the increment of acetate production as compared to the CAB1058 model. Whether ThIA replaced by AtoB is a good approach for butanol production?*

Response to reviewer 1: We do not agree with reviewer 1 on this point. The CAB1059 (AtoB replacing ThIA) possesses both a higher butanol to ethanol ratio but also a higher flux of butanol formation which is the only C4 compound produced by both strains and then indicating that AtoB is better than ThIA.

After properly addressing the above issues, I think the work is appropriate to be published in the journal.

Reviewer #2 (Remarks to the Author):

In this ms, Soucaille and collaborators carried out an extensive metabolic engineering of C. acetobutylicum to generate a strain that enables superior butanol/solvent titers and yields. They then integrated this strain into a continuous process that engages vacuum distillation to achieve an exceptional performance that exceeds all prior art and has all the characteristics of an economically attractive industrial process. The metabolic engineering work is tremendous, in its vision, design and execution, and exceeds all prior state of the art.

Overall, this is a superb achievement and story, and I enthusiastically recommend publication. Having said that, this current version is a very short format that one would suspect was submitted for publication to Nature or

Nature Biotechnology. As such, the story telling is too dense for most readers, especially for the non-specialist readers. I would therefore recommend that they recast the paper into the longer Nature Communications format, especially the metabolic engineering work. They can bring into the main ms a good deal of the now supplemental material.

Response to reviewer 2: As suspected by Reviewer 2, this work was initially submitted to Nature Biotechnology and asked to be transferred to Nature communication. The revised manuscript was modified to a longer version that we expect to be easier to read.

Reviewer #3 (Remarks to the Author):

The manuscript by Nguyen et al. describes an optimized process for microbial n-butanol production from glucose. The authors describe a rational approach for metabolic engineering of Clostridium acetobutylicum to minimize by-product formation and maximise the yield of n-butanol. Furthermore, they established a continuous production process employing this strain, which combines a higher cell density than usual with in-situ extraction of the toxic product by distillation. The obtained yield and productivity exceed values published up to now and could lead to a new economical exploitation of this historic process.

The details given appear appropriate, and reproducibility (incl. statistics) has been sufficiently shown in this reviewer's opinion. However, the manuscript itself is quite unpleasant to read as all genes are named with their abbreviations and most of them are not spelled out. It is recommendable to use the correct names of all enzymes – it might seem lengthy but it makes the text much more readable, particularly for readers who are not entirely in this field. It is also not appropriate to name microorganisms just with a number and not with their correct name. The English should be checked once more, there are some errors, such as "might be difficult to further overexpressed" or "its biochemical properties might also contributes" – there are more, which I do not list here.

Response to reviewer 3: The full names of the enzymes were already in the legend of figure 1 that presents the metabolic pathways. To take into account the comment of reviewer 3, we have also added the full names of the enzymes in the text of the main manuscript. On the other hand, as the recombinant strains have been extensively modified and their genotype is quite long, we did not name the different strains with their full name in the text but only in Table S2. English of the revised version has been re-checked by an editing company.

This reviewer disagrees with the simplified notion that the decline of the historic fermentation process was solely due to the development of petrochemistry. Clearly, petrochemistry played a significant role. However, also a drastic increase of the substrate costs was decisive. (Molasses were newly used for feed, which was much more profitable.) Other factors were involved as outlined in detail by Jones & Wood, who were cited by the authors. This factor appears important, because it will gain significance also nowadays, when more and more microbial processes start to compete for sugar.

Response to reviewer 3: The increase of substrate cost was now introduced as one of the factor explaining the decline of the fermentation process for butanol formation.

The last phrase notes that the new process would be economical. Could the authors give an estimate in numbers here? Why do they think so? Based on current glucose costs, which would be feasible n-butanol costs with their process? Ethanol for example is mostly not economical (outside of Brazil), when subsidies are taken out of the system, so this comparison makes it difficult to judge what the authors mean.

Response to reviewer 3: A company has made the economical evaluation of this new process but this information cannot be shared publically. But, to answer partially to reviewer 3, we can say that in the US the selling price of n-butanol is 2.3 times higher than ethanol (1.2 \$/kg versus 0.5\$/kg ISIS data from September 2017) while the yield of alcohol formation on glucose of our process is only 1.25 lower than the ethanol process! A sentence was added in the revised manuscript on this aspect.

Can the authors compare their technology to Gevo's technology for isobutanol production with yeasts? The Gevo technology is also based on in-situ removal of the isobutanol – the organisms is clearly completely different. This comment is not meant to lengthen the manuscript unduly – it would be just interesting to have a comparison to a (semi) commercial process, which is more closely related than ethanol production.

Response to reviewer 3: Unfortunately, to the best of our knowledge, the performances of the Gevo process are not public and it is then difficult to compare them.

REVIEWERS' COMMENTS:

Reviewer #1 (Remarks to the Author):

Lines 25, 48, 115 “n-butanol from glucose at the highest yield ever reported on a mineral media” This is probably true only for *C. acetobutylicum*, not for all organisms (Line 215). Higher butanol (>0.35 g/g) and ABE yields (0.47 g/g) have also been reported for *C. beijerinckii* BA101. This should be mentioned in the discussion so not to mislead the readers.

Line 39 – 35% better change to 0.35 g/g

Lines 138-142 “several methods of in situ butanol recovery including gas stripping, vacuum fermentation, pervaporation, liquid–liquid extraction, perstraction, and adsorption, have been investigated (27). However, none of them could either allow high volumetric productivities or could be easily scaled up at the industrial level. We then envisioned a continuous high cell density bioreactor with in situ extraction of the alcohols (n-butanol + ethanol) by distillation under low pressure...”

This statement is not true. Pervaporation has been used in both pilot scale and production plants for ABE fermentation. There are also many recent studies reporting the feasibility of gas stripping, liquid-liquid extraction, perstraction, and adsorption for in situ butanol recovery in ABE fermentation. In addition, the distillation under low pressure used in the present study is similar to the vacuum fermentation reported by Ezeji, which is not mentioned in the discussion. Only reference 27 is cited in the discussion, but it cannot support the authors’ claim that none of them could either allow high volumetric productivities or could be easily scaled up at the industrial level. Suggest to modify and tone down this sentence.

The high productivity achieved in the present study was because of the higher cell density maintained in the fed-batch fermentation. The authors should make a comparison of the specific cell productivity with other studies. Other in situ methods also could give high volumetric productivity if a high cell density were maintained in the system.

Table 1 – comparison is misleading as there are other studies with in situ butanol separation also gave comparable butanol yield, titer, and/or productivity but not listed in this table. Some of the other studies used *C. beijerinckii*, another solventogenic clostridia widely used in ABE fermentation. These studies should also be included in this table for a fair comparison.

Mixed use of period and common before the decimal point. Should use period for all values in the table.

Figure 3 – Same square symbols (with similarly light colors) are used for glucose and ethanol, which are difficult to be differentiated in printed B&W copy. Suggest to change to different symbols or use open instead of filled symbol for some of them for easier distinction.

Figure 4 – The dilution rates given in the diagram better be the average (steady state) values with

errors instead of a range so mass balance can be more easily done with the given information. It would be better to also give the volumes for the fermentor, boiler, and decanter. A more detailed figure caption or additional description should be given.

Reviewer #3 (Remarks to the Author):

I have no further comments.

Reviewer #1 (Remarks to the Author):

Lines 25, 48, 115 “n-butanol from glucose at the highest yield ever reported on a mineral media” This is probably true only for C. acetobutylicum, not for all organisms (Line 215). Higher butanol (>0.35 g/g) and ABE yields (0.47 g/g) have also been reported for C. beijerinckii BA101. This should be mentioned in the discussion so not to mislead the readers.

The results published by the group of Hans Blaschek with *C. beijerinckii* BA101 reporting a Butanol yield of 0.35 g/g associated to ABE yields of 0.47 g/g cannot be correct as the ABE yield reported is at least 20% higher than the maximum **theoretical yield** (0.4 g/g for an ABE ratio of 3:6:1). Furthermore, if a carbon balance is performed on these data there is a carbon excess in products of 25 % indicating that the yield were overestimated by 25%. This is the reason why we did not discuss these results in the manuscript.

Line 39 – 35% better change to 0.35 g/g
Change was made

Lines 138-142 “several methods of in situ butanol recovery including gas stripping, vacuum fermentation, pervaporation, liquid–liquid extraction, perstraction, and adsorption, have been investigated (27). However, none of them could either allow high volumetric productivities or could be easily scaled up at the industrial level. We then envisioned a continuous high cell density bioreactor with in situ extraction of the alcohols (n-butanol + ethanol) by distillation under low pressure...”

This statement is not true. Pervaporation has been used in both pilot scale and production plants for ABE fermentation. There are also many recent studies reporting the feasibility of gas stripping, liquid-liquid extraction, perstraction, and adsorption for in situ butanol recovery in ABE fermentation. In addition, the distillation under low pressure used in the present study is similar to the vacuum fermentation reported by Ezeji, which is not mentioned in the discussion. Only reference 27 is cited in the discussion, but it cannot support the authors' claim that none of them could either allow high volumetric productivities or could be easily scaled up at the industrial level. Suggest to modify and tone down this sentence.

The high productivity achieved in the present study was because of the higher cell density maintained in the fed-batch fermentation. The authors should make a comparison of the specific cell productivity with other studies. Other in situ methods also could give high volumetric productivity if a high cell density were maintained in the system.

We do not agree with reviewer #1 and we maintained that the other methods for the *in situ* butanol extraction cannot perform the rates of extraction obtained by distillation: in all the studies published the productivities were never higher than 2 g/l.h compared to 15 g/l.h in our study. Furthermore, we do not obtained high productivities in continuous culture (and not in Fed-batch as said by reviewer 1) because we have high cell density but the reverse because we could achieved high rate of butanol **extraction in situ** then we could maintain butanol concentration below inhibitory concentration and reach high cell density and productivities which others could not with the other methods of extraction they used. Finally the vacuum fermentation reported by Ezeji is also totally different from our study: this method does not provide selectivity for butanol removal (a lot of water is also removed from the fermentor) and a similar butanol titer to our study will never be obtained by this method. Furthermore the cost to maintain the vacuum will be very high due to the high amount of gas produced during the fermentation (that has to be pumped to maintain the fermentor under vacuum).

Table 1 – comparison is misleading as there are other studies with in situ butanol separation also gave comparable butanol yield, titer, and/or productivity but not listed in this table. Some of the other studies used C. beijerinckii, another solventogenic clostridia widely used in ABE fermentation. These studies should also be included in this table for a fair comparison.

We have now included the studies with *C. beijerinckii* with *in situ* butanol recovery. They also confirmed our comments made above that much lower productivities were obtained with other extraction methods.

Mixed use of period and common before the decimal point. Should use period for all values in the table.
Table was modified to use period for all values

Figure 3 – Same square symbols (with similarly light colors) are used for glucose and ethanol, which are difficult to differentiated in printed B&W copy. Suggest to change to different symbols or use open instead of filled symbol for some of them for easier distinction.

Figure 3 was modified to make it readable in printed B&W copy

Figure 4 – The dilution rates given in the diagram better be the average (steady state) values with errors instead of a range so mass balance can be more easily done with the given information. It would be better to also give the volumes for the fermentor, boiler, and decanter. A more detailed figure caption or additional description should be given.

Average with errors for dilution rates are now given. Volume of fermentor, boiler and decanter were given in material and methods but are now also included in the legend of the figure.